



# EBSD in Antarctic and Greenland Ice

Ilka Weikusat[1], Ernst-Jan N. Kuiper[1,2], Gill M. Pennock[2], Sepp Kipfstuhl[1], Martyn R. Drury[2]

[1] Alfred Wegener Institute for Polar and Marine Research, Am Alten Hafen 26, 27568 Bremerhaven, Germany

[2] Faculty of Earth Science, Utrecht University, Postbus 80021, 3508 TA Utrecht, The Netherlands

5  *Correspondence to*: Ilka Weikusat (ilka.weikusat@awi.de)

**Abstract.** EBSD provides information for the characterization of subgrain boundary types and dislocation activity during deformation. EBSD microstructure in combination with light microscopy measurements from ice core material from Antarctica (EPICA-DML deep ice core) and Greenland (NEEM deep ice core) are presented and interpreted regarding substructure identification and characterization. Electron backscattered diffraction (EBSD) analyses suggest that a large portion of edge dislocations with slip systems basal <a> gliding on the basal plane were involved in forming subgrain boundaries. However, an almost equal number of tilt subgrain boundaries developed, involving dislocations gliding on non-basal planes (prism <c> or prism <c+a> slip). A few subgrain boundaries involving prism <a> edge dislocation glide, as well as boundaries involving basal <a> twist dislocations, were also identified. The finding that subgrain boundaries occur, made up of dislocations gliding on non-basal planes, are as frequent as basal plane slip systems, is surprising. These findings are expected to have an impact on the discussion of rate-controlling processes for the ice flow descriptions of large ice masses with respect to sea-level evolution. For subgrain boundaries not related to the crystallography of the host grain alternative formation processes are discussed.

## 1 Introduction

20  Ice, particularly the extensive amounts found in the polar ice sheets, impacts directly on the global climate by changing the albedo and indirectly by supplying an enormous water reservoir that affects sea level change (IPCC 2014, Stocker et al. 2010, IPCC 2007, Lemke et al. 2007, Bindoff et al. 2007). The discharge of material into the oceans is partly controlled by the melt excess over snow accumulation and partly by the dynamic flow of ice (Hock et al. 2005). In addition, rapid transportation of ice towards coasts occurs by basal sliding over the bedrock, which includes heterogeneous and complex

25  sub-glacial processes (Vaughan et al. 2007, Hughes 2009), such as deformation of sediments or lubrication by melt water at





the base of ice sheets (Thoma et al. 2010, Beem et al. 2010, Wolovick et al. 2016). In particular, fast discharge by ice streams accounts for up to several hundreds of meters per year surface velocity of parts of the ice sheets (Joughin et al. 2015). However, the movement of material towards these rapid flow regions in ice streams and near to the base of the glacier occurs by internal deformation of the whole ice body. Internal deformation is therefore responsible for the convergent flow

geometries at the onset of ice streams (Bons et al. 2016), although at deformation rates of only a few cm to meters per year surface velocity, which are much slower than movement caused by ice stream flow.

Ice sheet flow models (e.g. Greve and Blatter 2009, Huybrechts 2007) are based on Glen's flow law (Glen 1955), which has a power-law form. This law was derived from experimental deformation of small ice specimens at much higher stresses (0.1 to 1 MPa) than that expected to occur in ice sheet flow (<0.1 MPa). Glen's law excludes the full suite of natural physical

processes that might also lead to a change in the dominant deformation mechanism, or the influence of second phases and impurities. Rate-limiting processes for different flow conditions are a matter of extensive discussion, and include contributions from various deformation mechanisms, the formation of a crystallographic preferred orientation (CPO), recrystallization and the occurrence of a liquid phase (Alley et al. 1992, Goldsby and Kohlstedt 1997; 2001, Duval et al. 2002, Alley et al. 2005, Song 2008, Schulson and Duval 2009).

The main deformation process in the creep deformation of natural ice is presumed to be intra-crystalline dislocation glide and climb (e.g. Schulson and Duval 2009, Faria 2014b). Knowledge of the dislocation activity and dislocation types is, therefore, of importance for a complete understanding of ice deformation. Previous studies that focused on descriptions of dislocation types in ice (Ahmad et al. 1986, Baker 1997, Baker 2002, Baker 2003, Fukuda et al. 1969, Higashi et al. 1988, Hondoh 2000, Louchet 2004, Montagnat et al. 2004a, Sinha et al. 1978, Whitworth et al. 1978, Breton et al. 2016) are

mainly based on experimental deformation carried out under laboratory conditions (Montagnat et al. 2015) on polycrystalline (Barrette et al. 1994, Bryant and Mason 1960, Wei and Dempsey 1994) and/or single crystal ice (Montagnat et al. 2003, Montagnat et al. 2001). To understand the deformation in a naturally occurring ice sheet, analysis of dislocation activity is needed of large sections of the naturally occurring ice body, preferably along the length of an ice core. This study presents the first step in an ongoing study and presents dislocation types from a single depth.

Ice found on Earth is hexagonal and dislocation activity along basal (0001) versus prismatic {h0-i0} and pyramidal {h0-il} planes is highly anisotropic. Single crystal deformation tests have shown that critical resolved shear stresses on non-basal planes require 60 to 100 times higher stresses than activating dislocation glide on basal planes (Duval et al. 1983, Ashby and Duval 1985), so slip on basal planes is expected to dominate. This comparative ease of basal slip is possibly caused by dissociation of dislocations into partial dislocations. Dissociation may occur by dislocations that extend in the basal plane

and are connected by short segments that glide on non-basal planes (Fukuda et al. 1987, Hondoh 2000, 2010, Breton et al.

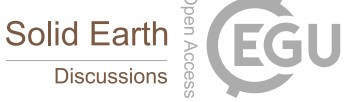

2016). The splitting of dislocations into partials is possible because of a very low stacking fault energy on the basal plane (Silva and de Koning, 2012) and related to the similarity of hexagonal and cubic ice structures (Hondoh 2015). However, non-basal slip has been observed by X-ray tomography in ice single crystals (e.g. Fukuda 1987, Higashi 1988, Baker 2003), although only as short segments that connect the partial dislocations in the basal plane. These short segments provide

multiplication mechanisms and accommodate strain heterogeneities, but are not expected to contribute to the dominant macroscopic deformation, despite their rapid migration rate (Shearwood1991).

Microstructure mapping using light microscopy (LM) of large area (10 x 10 cm) etched surfaces (Figure 1), cut parallel to the core axis (Figure 1b), was developed to study ice core samples (Kipfstuhl et al. 2006). The procedure was carried out on-site shortly after drilling of the core to avoid and in later measurements recognize relaxation effects that occur during

standard sample storage at temperatures of -10 to -30°C (Miyamoto et al. 2009, Weikusat et al. 2012). During recovery, many dislocations polygonise into subgrain boundaries (SGBs). Grain boundaries (GBs) and SGBs in ice can be differentiated using LM images by the depth of thermally etched grooves (Mullins 1957, Nishida and Narita 1996, Arnaud et al. 1998, Barnes 2003). SGBs have lower misorientations than GBs, so form shallower etch grooves (Saylor and Rohrer 1999), which have lower contrast than GBs in LM images and appear grey rather than black (Figure 1a). Advanced

automatic image processing tools were used to extract and parameterize the GBs and SGBs (Binder et al. 2013b) to give grain size and grain shape statistics (Weikusat et al. 2017). This information was used to estimate driving forces for grain boundary migration along deep ice cores (Binder et al. 2013a). In addition to contrast differences in the etched boundaries, the shape of the boundary was used to distinguish between SGBs and GBs: SGBs generally have straighter segments that are controlled by recovery of dislocations and locally tend to fade out, whereas GBs tend to be smoothly curved and are

continuous.

X-ray Laue diffraction (Weikusat et al. 2011) confirmed that the less deeply etched boundaries observed in LM (Weikusat et al. 2009b) all had low misorientations <5º and were SGBs. The SGBs in natural polar ice (Weikusat et al. 2009, Kipfstuhl et al. 2009), as well as in experimentally deformed artificial ice (Hamann et al. 2007, Weikusat et al. 2009a), can be grouped into certain types using LM. These SGB types are characterized according to the alignment of their intersection line (trace)

with the etched surface. In polarised LM, the c-axis of the ice is known, so the SGB trace can easily be described with respect to the c-axis, or basal plane. Four SGB types were identified (Figure 1c, Weikusat et al. 2009b): the N-type SGB has a trace that is predominantly normal to the basal plane, the P-type has a trace parallel to the basal plane, Z-type has an irregular zigzag trace, where one orientation dominates which is predominantly normal to the basal plane, and lastly, a SGB with no strong alignment to the basal plane which is often curved.



The N-type SGB is also called the "Nakaya"-type, after the first extensive description in 1958 by Nakaya of low misorientation angle boundaries in ice occurring "in the prism plane ... perpendicular to the gliding layers" (Nakaya 1958). This type of SGB resembles the classical perception that glaciologists have of an ice grain undergoing recovery and polygonization of dislocations into a SGB (for example, Figure 2 in Alley et al. 1995). The N-type SGB is usually deeply

etched and cuts across the whole, or a large part, of the grain. Several N-type SGBs can cluster together with one or two fainter, shorter sub-parallel SGBs. The P-type SGBs are also straight and often occur in parallel swarms that typically do not cross a grain, but fade out after a short distance. Z-type SGB often appear in networks and intricate patterns that usually form short lengths that do not completely cross a grain. Z-type SGBs are typically less deeply etched and disappear towards the core of the grain. The curved SGB types are rarely observed, possibly because the surface orientation influences the

sublimation (see method explanation in Weikusat et al. 2009).

Based on analysis of LM mapping along an ice core, the P-type SGB is the most frequent of all subgrain boundary types at all depths of the EDML ice core, followed by Z-type, N-type and curved types (Weikusat et al. 2009). X-ray Laue diffraction (Weikusat et al. 2011) of approximately 240 shallow sublimation grooves revealed misorientations of less than 5°. Many of them (30-40%) were below the angular resolution of 0.5° for Laue, while the majority (50-60%) were below 3°. Only a few

percent had misorientations between 3 and 5°. From those SGB found with >0.5° (165) 45 indicate that the formation of SGB involved significant activity of non-basal slip (Weikusat et al. 2011). Although X-ray Laue diffraction is very useful technique for determining the slip system of a boundary of large-grain size, polycrystalline ice, measurements and processing of Laue patterns are semi-automatic (Miyamoto et al. 2011) and thus time consuming and only a limited number of boundaries could be measured. Identifying the proportion of non-basal dislocation activity is crucial for understanding the

deformability of ice. Significant non-basal slip might cause a change in the deformation rate-controlling processes, and lead to a change in the stress exponent from a value of three at low stresses towards two at higher stresses (Montagnat and Duval 2000). To determine slip systems in a large number of SGBs, electron backscattered diffraction (EBSD) is needed.

Cryo-EBSD has been successfully used to study ice (Iliescu et al. 2014, Obbard et al. 2006). Maintaining a stable ice surface during EBSD assessment (Weikusat et al. 2011, Prior et al. 2015) is essential for these studies to allow correlation between

LM etched surfaces and also to differentiate between in-situ deformation occurring in the ice sheet and any relaxation effects that occur during the various sample preparation procedures (Weikusat et al. 2010). When a subgrain boundary is formed by the polygonisation of geometrically necessary dislocations (Humphreys and Hatherly 2004, Weertman and Weertman 1992, Hirth and Lothe 1982), the crystallography (boundary plane and rotation axis) of subgrain boundaries can be interpreted in terms of the active dislocation slip systems. EBSD provides full crystallographic information (Randle and Engler 2000) from

which slip systems can be derived (Trepied 1980, Neumann 2000, Lloyd et al. 1997, Prior et al. 2002, Piazolo et al. 2008,



Weikusat et al. 2011, Montagnat et al. 2015), although SGB plane traces are best obtained from the corresponding LM images to avoid any artefacts (Weikusat et al. 2011).

In this work we assess SGB types in two ice core samples from Antarctica and from Greenland using LM and cryo-EBSD. We assess a single depth in each ice core. The boundary types are described using the same terminology as that used in our earlier LM studies. The slip systems from a significant number of boundaries in natural ice are determined using EBSD, so that we can verify the importance of non-basal slip activity that were previously identified in a limited number of boundaries using combined X-ray and LM studies.

## 2 Material and Methods

Samples from two polar ice cores were used for this study: NEEM (North-Greenland Eemian ice drilling) and EDML (European project for ice coring in Antarctica - Dronning Maudland). We collected data on a single, roughly equivalent, depth for each core, in order to obtain a first comparable statistic of SGB type.

The EDML samples were obtained from 656 m depth of the EPICA deep ice core, drilled between 2003 to 2006 at Kohnen station (75°00'S, 0°40'E, 2892 m a.s.l.) in the Atlantic sector of East Antarctica (Oerter et al. 2009, Ruth et al. 2007, Wilhelms et al. 2007, Wilhelms et al. 2014). The NEEM samples were obtained from 719 m depth and were drilled between 2008 and 2012 in North-West Greenland (77°27'N, 51°03'W, 2484 m a.s.l.) (NEEM 2013, Rasmussen et al. 2013).

Both drill sites are located on ice ridges or ice divides with horizontal flow velocities at the surface of 0.8 m/a for EDML (Wesche et al. 2007) and 6 m/a for NEEM (personal comm. Christine Hvidberg, Okt. 2016). The divide flow at EDML is significantly more divergent compared to NEEM (Weikusat et al. 2017). This flow results in a stronger and clearer c-axes preferred orientation (CPO), giving a vertical girdle in pole figures in the central part of the core, while in NEEM the c-axis CPO resembles an elongated single maximum (Montagnat et al. 2014).

In order to compare similar flow conditions, we sampled from depths that reflect the deformation regime expected at ice divides, that is, extension normal to the dividing ridge, with almost no deformation along the ridge, and with vertical compression (tri-axial deformation state). The c-axis CPO was not strongly developed for the depths selected. Both sample



depths originate from snow deposited during the Holocene (Ruth et al. 2007, Rasmussen et al. 2013), and have comparably small impurity concentrations (e.g. Dansgaard et al., 1982, EPICA 2004, 2006, Kuramoto et al. 2011, Wegner et al. 2015).

Samples were transported at -25°C from Antarctica to Bremerhaven, and from Greenland to Bremerhaven via Copenhagen and subsequently stored at -30°C. Samples (50 x 100 x 90 mm) were cut parallel to the long axis of the drill core (Figure 1b)
using a band saw and polished in a -25°C cold laboratory in Bremerhaven according to standard procedures using a microtome (e.g. Pauer et al. 1999, Wang et al. 2002).

LM microstructure mapping was performed at -20°C at AWI to obtain an overview of the microstructure using a Leica DMLM (Kipfstuhl et al. 2006) for high spatial resolution images (1 pixel edge 3.3µm) and a large area scanning macroscope (Krischke et al. 2015, 2015a) with slightly lower spatial resolution (1 pixel edge 5µm). The method was also applied on-site
immediately after drilling (in 2003 for the EDML and in 2010 for the NEEM samples), which allowed us to monitor any microstructural changes caused by relaxation, by pressure and temperature change during transportation and storage and to locate regions of interest for EBSD, particularly as the subgrain microstructures were heterogeneously distributed.

Typical SGB microstructures were identified in LM images and selected for cryo-EBSD analysis. Small specimen blocks (~8 x 8 x 5 mm) were cut using a handsaw for EBSD studies. As SGBs in ice are mainly located close to GBs, triple junctions or
grain necks (Weikusat et al. 2009), EBSD mapped areas could be selected such that up to ~15 SGBs could be measured in each map and up to 10 maps could cover almost the total area of the small block. The small blocks were again polished by microtoming and sublimation. A further high-resolution LM microstructure mapping step was performed. Polished specimens were stored in solid carbon dioxide (dry ice) at about -70°C to reduce any further sublimation during transfer to Utrecht University (about 2 days to 3 weeks). A total of 4 samples were examined for the NEEM core and 10 for the EDML
core.

Data was collected using an FEI Nova Nanolab 600 scanning electron microscope (SEM), equipped with an EBSD detector (Oxford Instruments HKL Technology, Abingdon, U.K.), a cryo-preparation station and a cryo-stage (Quorum Technologies Ltd, Ringmer, U.K.). Samples transfer to the SEM involved a further short sublimation step of a few minutes under vacuum (Weikusat et al. 2010). Typical measurement conditions for EBSD mapping were a working distance of 6 – 8 mm, 123 to
150°C sample stage temperature (-145 to 170°C cold trap temperature), $5 \times 10^{-5}$ to $3 \times 10^{-6}$ hPa chamber pressure, 10 or 20 kV accelerating voltage and 8.4 nA beam current. The sample surface was kept stable for 8-9 h under these conditions. Microstructures were imaged using secondary electrons. Channel 5 software (Oxford Instruments) was used to collect and analyse the EBSD data. Typical mapping rates were 0.15s per pixel and indexing rate 90%. EBSD data processing was



performed using standard noise reduction and orientation filtering as described by (Weikusat et al. 2010). Angular resolution was about 0.5° after orientation averaging. For EDML (656 m depth) four individual samples were mapped with EBSD maps were measured; for NEEM (719 m depth) ten individual samples were mapped EBSD.

Each individual SGB was imaged using LM and relocated in the SEM before EBSD mapping. The pixels across boundaries in an EBSD map were selected manually and saved as subsets of the map, so that a narrow range of orientations from the boundary were analysed, which avoided any orientation changes occurring away from the subgrain boundary. An upper misorientation angle of 5° was taken for all SGBs (Weikusat et al 2011). Neighbour pixel misorientations between 0.5 and 5° were plotted in an inverse pole figure (IPF) for each subset to determine the rotation axes of the SGBs. The trace of the SGB was determined from the LM image and compared to the EBSD orientation information to avoid any artefacts caused by charging or poor sample alignment with respect to the tilt axis (Weikusat et al. 2010).

# 3 Results

## 3.1 Subgrain boundary types

Typical LM images for the EDML and NEEM samples are shown in Figures 2 and 3 respectively. (Figures 2a and 3a). The stripy contrast in Figure 2 (an artefact of automatic image processing) is aligned to the ice core axis, as in Figure 1. Two SGB types are labelled in Figure 2a: a P-type boundary, with a trace parallel to the basal plane and an N-type boundary, with a trace that is predominantly perpendicular to the basal plane.



## 3.2 Misorientation angle range used for subgrain boundaries using EBSD

Approximately 230 individual SGBs were measured in the EDML samples and ~180 in the NEEM samples using EBSD. More SGBs (ca. 30% for EDML and 20% for NEEM) were identified in LM images than in EBSD maps. Red arrows show three such boundaries in an LM images (Figure 2a and 3a) that were absent in the corresponding EBSD maps (Figure 2b and

3b) of the same area. The boundaries are likely SGBs with misorientations that were below the angular detection limit of ~0.5 to 0.7° found using our EBSD mapping conditions.

## 3.3 Analysis of subgrain boundary types

The same morphological SGB types (N-, P-, Z-type and curved) identified in LM images were observed in EBSD mapped microstructures. The different SGB types were analysed using LM images and EBSD mapped data (Figures 2 and 3): the

trace of the boundary and the rotation axes, R, were used to determine the slip systems.

### 3.3.1 Subgrain boundary traces

Small rotation differences were observed between many EBSD images and the LM images (see Figures 2a and 2b). The orientations of the SGB traces were therefore made using the LM images, combined with EBSD orientation data (Figures 2c and 3c).

### 3.3.2 Subgrain boundary rotation axes

12 SGB subsets used to determine rotation axes are highlighted in both Figure 2 for the EDML ice sample and in Figure 3 for the NEEM ice core sample. The axes are displayed in inverse pole figures (Figures 2d and 3d). The rotation axes are spread over a large portion of the IPF. Nevertheless, some rotation axes cluster around certain poles.

The following types of rotation axes, R, found in the two samples are summarised below.

20    1.      R is in the basal plane (0001) with a general direction <hki0> (Figure 2d, SGBs 1,5,6,7 and Figure 3d SGBs 6,11,12). Two special cases of this type of rotation axes occur, where R is either parallel to the prism plane normal, <01-10> (Figure 2d, SGB 2,3 and and Figure 3d, SGBs 5,8), or parallel to the a-axis (<1-210>, Figure 2d, SGB 4,10 and Figure 3d, SGBs 2,7,9).

2.      R is parallel to the c-axis, <0001> (Figure 2d, SGB 8 and Figure 3d, SGB 3).

25    3.      R is parallel to the pyramidal plane normal, <h0-il> (Figure 2d, SGB 9 and Figure 3d, SGB 4).

4.      R is not parallel to a specific orientation but dispersed (Figure 3d, SGBs 10,11,12 and Figure 4d, SGB 1).



### 3.3.3 Combination of subgrain boundary traces and rotation axes

In order to make a comparison with SGB statistics used in LM studies, we choose similar arrangements of the SGB plane with respect to the basal plane. These arrangements are shown in Figure 4, an extended version of that given by Weikusat et al. (2010, 2011). A SGB boundary plane that is perpendicular to the basal plane (N-type) is shown in Figures 4a and 4b and a

SGB plane that is parallel to the basal plane (P-type) is shown in Figures 4c and 4d.  For each of these arrangements, simple end member arrangements are chosen for the rotation axes (dots in Figure 4e) with only one end member link R lying in the basal plane (shaded in Figure 4e). In Figures 4a and 4d, R is in the basal plane (perpendicular to the c-axis) and in Figure 4b and 4c R is parallel to the c-axis.  A shorthand notation is introduced to describe these four types of boundary, N[a], N[c] P[c] and P[a] (Table 1). N[a] describes a SGB trace normal to the basal plane (N-type) with rotation axis in the basal plane,

(e.g. <a>, or <01-10> or <hk-i0>), this is a tilt SGB (Figure 4a). N[c] is also a tilt SGB of N-type but with a rotation axis parallel to the c-axis (Figure 4b). P[c] has a SGB parallel to the basal plane (P-type) with rotation axis [c] and is a twist type of boundary (Figure 4c). P[a] is also a tilt boundary with the boundary parallel to the basal plane and a rotation axis in the basal plane, (e.g. <a>, or <01-10> or <hk-i0>).

A compilation of the complete data set of combined information from rotation axes and SGB trace alignments is given in

Table 2. The number frequency of SGBs is shown in terms of the rotation axes and the orientation of the SGB plane trace with respect to the basal plane trace.

The majority of SGBs could be allocated to certain classes with a well-defined trace orientation and rotation axis. The most common SGB types have R in the basal plane, with roughly an equal proportion of P and N plane traces, and R normal to the

basal plane. SGBs of the P[a] type are shown in Figure 2 for SGBs 5 and 7 and in Figure 3 for SGBs 7, 9 and 11.  SGBs of the N[a] type are shown in Figure 2 for SGB 1-4 and in Figure 3 for SGBs 2, 5, 6, 8 and 10. SGB 8 in Figure 2 and SGB 3 in Figure 3 are examples of a P-type SGB with a rotation axis [0001] (P[c] type). SGBs with this rotation axis are rare in both ice cores, particularly the NEEM core (Table 2). SGB 3 in Figure 3 is another example of a SGB with R=[0001], however combined with a N-type SGB trace (N[a]). SGB 9 (Figure 2) and the left half of SGB 4 (Figure 3) are examples of a very

rare N-type with R <h0-il>. SGB 10 and 11 (Figure 2) show N-type trace with R dispersed: there is a slightly higher fraction of dispersed axes in EDML compared to NEEM.

The EBSD data are grouped (Table 2, rightmost 3 columns) into simplified statistics in order to make a direct comparison between EBSD and X-ray Laue diffraction data published for the EDML ice core (Table 2 in Weikusat et al. 2011). Table 2

confirms the conclusion of the X-ray Laue study, which showed a dominance of SGB with a rotation axes in the basal plane.



## 4 Discussion

### 4.1 Comparison with light-microscopy and X-ray Laue diffraction results

A comparison of LM microstructure maps with high-resolution crystal orientation measurements reveals information on the
threshold misorientation for subgrain boundary towards grain boundary transition. First results for this threshold were
reported with X-ray Laue measurements on Antarctic ice (Weikusat et al. 2011). SGB below the relative angular resolution
(ca. 0.5°) are still depicted by LM sublimation etching. The subgrain boundary to grain boundary transition is characterized
by the change of the boundaries' properties (e.g. Gottstein 1999), which are related to the boundaries intrinsic energies. One
of these boundaries' properties is the sublimation behaviour, which is directly used in our LM method (Kipfstuhl et al. 2006,
Weikusat et al. 2011). Measurements of grain boundary energies in ice are rare and only possible with relative measurements
on symmetric tilt boundaries (Boinovich and Emelyanenko 2014, Suzuki 1972, Suzuki 1970, Ketcham 1969). However,
these results reveal the largest increase of energy between 0° to below 10°. This is in accordance with the high sensitivity of
the sublimation etching method used in LM microstructure mapping: SGB of lowest misorientation, thus lowest energies
reveal the faintest, however still visible sublimation grooves. For this EBSD study we adopted the threshold misorientation
measured by X-ray Laue between SGB and GB (3-5°) for Antarctic and Greenlandic ice, which is low compared to rock-
forming minerals (5-10°) and metals (15°).

Our results confirm that subgrain boundaries are very common in ice, such that the majority of all grains (60-80%) in all
depths in Antarctica and Greenland ice develop them (Weikusat et al. 2009, Knotters 2016). This high fraction may be due to
a highly efficient recovery, where dislocations walls and subgrain boundaries form. This efficiency is possibly related to the
high temperatures (>0.8 Tm) of natural ice deformation. The frequent observation of SGB in our samples is in accordance
with classical experiments, which bend a single crystal over supporting wedges. Nakaya (1958) observed sharp bends
forming subgrain boundaries with only few minutes of arc in these experiments.



### 4.2 Subgrain boundaries controlled by host grain crystallography

Those SGBs with clustered rotation axes were described in terms of the host crystallography. The spread in orientation of these axes is consistent with errors expected from low angle misorientation boundaries (Prior et al. 1999). Our study provides statistics of around 400 SGB in Antarctic and Greenland ice. Analysis of LM and EBSD microstructures are

described in the following paragraphs in terms of the possible slip involved in forming the observed SGB types. By means of the geometrical relation with the host crystal the SGB types are interpreted in terms of active dislocation slip systems.

*N-type tilt boundaries with rotation axis in basal plane – N[a]* A subgrain boundary normal to the basal plane (N- or z-type) and a rotation axis R in the basal plane with R=<hki0> can be produced by an array of Burgers vector b=<a> edge dislocations gliding in the basal plane (Figure 4a). The observed rotation axes of N-type boundaries are often a general

orientation <hki0> or sub-parallel to <01-10> or <1-210>, which can be explained by the occurrence of dislocations with Burgers vector of [a1] and [a2]. The vast majority of all N-type SGBs (ca. 85 to 75%) have rotation axes in the basal plane and thus are tilt boundaries made up by b=<a> edge dislocations that can glide on the basal plane (Table 2).

*N-type tilt boundaries with rotation axis parallel to c-axis – N[c]* A boundary with a plane normal to the basal plane and with R=[0001] can be produced by an array of b=<a> edge dislocations on a prismatic glide plane such as {10-10} or {1-

210}. These are dislocations with a Burgers vector in the basal plane, but with non-basal glide planes (Figure 4b). This type has been observed in few examples only making few percent of the N-type SGBs (Table 2).

*P-type twist boundaries – P[c]* A subgrain boundary accumulating dislocations parallel to the basal plane (P-type) and a resulting rotation around the c-axis [0001] can be produced by two or three sets of basal screw dislocations (Figure 4c) with Burgers vectors b=<a> (Hondoh 2000). Basal twist boundaries represent a minority (~2 to 8%) of all parallel SGBs (Table

20   2).

*P-type tilt boundaries – P[a]* A dislocation wall parallel to the basal plane (P-type) and a resulting rotation in the basal plane (<-12-10> or <01-10>) can be produced by an array of non-basal edge dislocations (Figure 4d). An array of b=[c] edge dislocations with {1-100} glide plane has a rotation axis R=<1-210>, while an array of b=[c] edge dislocations with {1-210} glide plane has R=<10-10>. Many boundaries have a general rotation axis <hki0> suggesting that the glide planes of the

b=[c] dislocations can occur on a range of {hki0} prismatic planes. Dislocation arrays of b=<c+a> with two Burgers vectors, b1=c+a1 and b2=c+a2, could also produce this type of SGB. Most of all P-type boundaries (~72 to 90%) are tilt boundaries with non-basal dislocations (Table 2).

A main result of our study is that a very high fraction of SGBs in polar ice (ca. 30-40% of all SGB) consist of non-basal dislocations and so could be formed by non-basal slip. Although somewhat unexpected, this result is in accordance with

previous, rare measurements (Weikusat et al. 2011). According to Hondoh (2000) the strain produced by non-basal slip is



rather limited and in the case of <c+a> pyramidal slip, limited strain can result in the formation of an N-type tilt wall made up of immobile [c] dislocations. In contrast, only a small proportion of the dislocations involved in basal slip, are likely to become organized into dislocation walls. The activation of non-basal slip suggests that locally within grains stresses were high enough to activate the hard slip systems. In addition, we can also conclude that deformation did not occur via basal slip

with strain incompatibilities between grains accommodated only by grain boundary processes such as sliding (Goldsby and Kohlstedt 2001) and/or grain boundary migration (Montagnat et al. 2002).

Many studies assume that all subgrains form by polygonization of geometrically necessary dislocations (Ashby 1970, Ashby and Duval 1985), however, in-situ deformation experiments on rock analogues (Means and Ree 1988) as well as observations in some rock-forming minerals (Drury and Pennock 2007) have shown that subgrains may originate in several

ways. Means and Ree (1988) recognised seven types (I to VII) of subgrain boundary in octachloropropane (OCP) deformed at 0.7 to 0.8 Tm. The crystallography of Means and Ree type I SGBs are determined by the geometrically necessary dislocations that accommodated the lattice bending in the grain leading to polygonization. Means and Ree (1988) suggested that their type II SGBs formed by glide polygonization, and thus also reflect the active slip systems in the grain. Type II, however, will experience some modification of the SGB crystallography, as these are mobile tilt boundaries and a boundary

migrating through a bent lattice will collect all dislocations. Means and Ree (1988) found that two thirds of the boundaries in their OCP experiment were type I and II boundaries, which suggests that the crystallography of SGBs is strongly influenced by the active slip systems.

### 4.3 Subgrain boundaries without connection to host grain crystallography

The geometry of approximately 20-30% of all SGB cannot be related to the host crystal orientation (Table 2). These SGB

could be formed by a processes similar to that described by Means and Ree for type III to VII. The crystallography of type III and IV SGBs should have no simple relationship to the active slip systems, because they are formed by misorientation reduction at grain boundaries and by impingement of migrating boundaries. These types of SGBs could occur in polar ice, with its characteristic strong CPO development with ice sheet depth (Weikusat et al. 2017, Jansen et al. 2016, Montagnat et al. 2014, Fitzpatrick et al. 2014, Faria et al. 2014a and references therein). No particular rotation axis is expected for these

boundaries, although the occurrence of a strong CPO will tend to produce preferred geometries, controlled by the type of CPO (Mainprice et al. 1993). SGBs that develop from GBs may be recognisable as such because they are part of the grain boundary network, whereas the vast majority of SGB are internal structures that occur inside grains and are not part of the GB network. This is the case for many natural ice samples, not only for the samples from depths presented here (Weikusat et al. 2009a,b). Means and Ree-type V SGB formed by impingement of two migrating SGBs and will develop misorientation

that is the sum of the two boundaries. If the boundaries are formed by the same slip system, then only the angle will be changed. Impingement of boundaries formed by different slip systems, will produce a new misorientation, that combines the dislocation content of the two boundaries. Such SGB can be described as mixed boundaries composed of a mix of

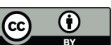


dislocations. The type VI SGBs grow behind a migrating grain boundary, so their length does not reflect slip system activity; however, such boundaries would grow from initial boundaries produced by deformation, so the boundary misorientations should be related to the active slip systems. Given the extensive grain boundary migration that occurs at all levels in ice-cores (Montagnat and Duval 2004, Weikusat et al. 2009a), it is likely that many SGBs have been extended by growth behind

migrating boundaries. This is conceivable in particular with the common observation that only a part of SGBs crosses grains completely while many of them fade out towards the centre of a grain (Weikusat et al. 2009b, Hamann et al. 2007). As Means and Ree-type VI originate from mobile grain boundaries, this process can likely produce SGBs showing curved shapes as observed for many SGB in the trace class "other" (Table 2). Type VII SGBs evolve statistically from an unbent lattice in Means and Ree deformation tests.

## 10  4.4 Special subgrain boundaries in ice

For the ice-core samples from EDML and NEEM, the common N[a] type SGBs (Table 2) are made of basal dislocations (basal <a> slip), while the equally common P[a] boundaries consist of non-basal b=[c] or <c+a> dislocations (prism [c] slip). There is a tendency for SGBs to be more strongly visible near grain boundaries (Weikusat et al. 2009, Faria et al. 2014b). The same type of SGBs occur in experimentally deformed ice (Hamann et al. 2007), which had a fine initial grain size and

underwent significant grain growth during deformation. The SGBs were concentrated near the grain boundaries and appeared to pin the migrating grain boundaries. This shows that the P- and N-type boundaries in ice could be partly extended by growth behind migrating boundaries such as the Means and Ree-type VI. Alternatively, it is common in many materials to develop a core and mantle substructure (Gifkins 1976, Faria et al. 2014b) with more SGBs with higher misorientations near the grain boundaries. The development of non-basal SGBs in the grain mantle is consistent with the

stress concentration near grain boundaries, as has been found for hexagonal metals (Ion et al. 1982, Drury et al. 1985). Hondoh (2000, 2010) proposed that P[a] boundaries could be formed by local slip of <c+a> screw dislocations in grains that were shortened parallel to the c-axis. The local slip of the <c+a> screws formed an immobile edge dislocation dipole (Figure 5a). The dipole dislocations can dissociate into two dislocations, one with b=<a>, glides away leaving behind a b=[c] dislocation (Figure 5b). By repeating this process, an array of b=[c] dislocations in the basal plane, is formed (Figure 5c).

Strains perpendicular to the c-axis are accommodated by the local pyramidal slip and by the associated basal slip. Hondoh (2000, 2010) also suggested that strains would be accommodated by climb of the b=[c] dislocations. The other type of subgrain that indicates non-basal slip activity is the N[c] type (prism <a> slip), which occur but are uncommon in the ice core samples (Table 2). Hondoh (2000, 2010) suggested that the b=<a> dislocations gliding on the prismatic planes could also move on the basal plane, producing very irregular slip planes. The geometry of slip lines produced by prism <a> slip on

single crystals is similar to the Z-type SGBs (Weikusat et al. 2009), thus some Z-boundaries may be non-basal slip bands rather than subgrains.

The activation of non-basal slip has been directly observed in ice single crystals using X-ray tomography (Higashi 1988). Stress concentrations produced at grain boundaries or free crystal surfaces can result in the activation of non-basal dislocations (Wei and Dempsey 1994, Shearwood and Whitworth 1991, Levi et al. 1965). These studies show that strain accommodated by non-basal slip is limited as the non-basal dislocations only glide on small portions of the slip plane,

leaving behind a high density of immobile dislocations. As deformation in ice sheets occurs at very low stress and slow strain rates the occurrence of recovery could make non-basal slip easier (Faria et al. 2014b), although it should be noted that non-basal slip systems have a higher stress exponent than basal slip, so that the strength difference between the slip systems will increase with decreasing stress.

Recent numerical simulations of ice deformation (Llorens et al. 2016a,b) investigated the development of microstructure, subgrain formation and slip system activity during pure and simple shear deformation. In the simulations non-basal slip was 20 times harder than basal slip. The simulations showed that in pure shear, the activity of non-basal slip increased with strain, related to the development of a strong CPO, which made basal slip more difficult. Even at low strains, 20% of the slip activity was accommodated by non-basal slip in the simulations. The microstructures produced in the simulations show

heterogeneous activity of basal and non-basal slip systems, with different slip activity in different parts of grains resulting in formation of SGBs with a non-basal dislocation content (Llorens et al. 2017). The simulations show that a significant fraction of non-basal subgrains can be formed, with a relatively small activity of the harder non-basal slip systems.

## 5 Conclusions

We presented an analysis of SGB and their slip systems for 400 individual boundaries in Antarctic and Greenlandic deep ice cores (656 m and 719 m depth). Our study confirms the frequent occurrence of SGB in natural ice.

Our analyses of the SGB traces with respect to the crystal orientation and the rotation axis associated with the boundary revealed (1) subgrain boundaries that can be related to slip system activity by their geometric relation to the host crystal orientations (ca. 70-80%) and (2) subgrain boundaries that cannot be easily related to slip system activity (ca. 20-30%).

The slip systems interpreted from our subgrain boundary measurements show basal <a> slip is the most commonly observed slip system that form subgrain boundaries. However, almost equally occurrence of prism <c> or prism <c+a> slip is found, which can be further distinguished into primary prism <c> slip on {1-100} planes and secondary prism <c> slip on {1-210}

planes. Far less frequent are subgrain boundaries formed by screw dislocations with basal <a> slip (screw) and edge dislocation with prism <a> slip.

A main result of our study is that a very high fraction of SGBs in polar ice (ca. 30-40% of all SGB) consist of non-basal dislocations and so could be formed by non-basal slip. We conclude from the relatively high frequency of SGBs consisting

of non-basal dislocations that limited non-basal slip was active at these EDML and NEEM depths. This finding has important relevance for the discussion on rate-limiting processes for different flow conditions of ice, with respect to large scale ice modelling and its constitutive ice flow description.

The relevance of the extension of dislocations in the basal plane in ice for our observations is discussed and a new model of SGB formation by dipole separation is suggested.

Other possible means of producing subgrain boundaries in ice that cannot easily be related to slip system activity are discussed.  SGB originating from misorientation reduction of grain boundaries are likely to arise from the strong CPO development in natural ice. Growth behind migrating grain boundaries and core and mantle structures due to stress concentrations at grain boundaries are further likely SGB formation processes.

**Acknowledgements**

This project was funded by the German Science Foundation (DFG HA 5675/1-1 and WE 4695/1-2) within the SPP 1158 and the Helmholtz Association (VH-NG-802). The FIB-SEM microscope at University Utrecht is funded by NWO Groot and FEI. NEEM is directed and organized by the Center of Ice and Climate at the Niels Bohr Institute and US NSF, Office of Polar Programs. It is supported by funding agencies and institutions in Belgium (FNRS-CFB and FWO), Canada

(NRCan/GSC), China (CAS), Denmark (FIST), France (IPEV, CNRS/INSU, CEA and ANR), Germany (AWI), Iceland (RannIs), Japan (NIPR), Korea (KOPRI), The Netherlands (NWO/ALW), Sweden (VR), Switzerland (SNF), United Kingdom (NERC) and the USA (US NSF, Office of Polar Programs). This work is a contribution to the European Project for Ice Coring in Antarctica (EPICA), a joint European Science Foundation/European Commission scientific programme, funded by the EU and by national contributions from Belgium, Denmark, France, Germany, Italy, the Netherlands, Norway,





Sweden, Switzerland and the United Kingdom. The main logistic support was provided by IPEV and PNRA (at Dome C) and AWI (at Dronning Maud Land).

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



**Table 1: Summary and terminology of slip systems interpreted from analyses of LM and EBSD. R is the rotation axis of the subgrain boundary, <b> is the Burgers vector for slip. Prim = primary. Sec = secondary.**

| SGB geometry | | | Slip system | | Tilt/Twist | Frequency | Examples | |
|---|---|---|---|---|---|---|---|---|
| Name | Trace | Rotation axis <R> | Slip plane | Burgers vector <b> | | % of all SGB | EDML (Fig. 3) | NEEM (Fig. 4) |
| **N[a]** | N-type | <a> or <hki0> | Basal | <a> | Tilt | 27-41 | SGBs 1,2,3,4 | SGBs 2,5,6,8,10 |
| **N[c]** | N-type | [c] | Prism | <a> | Tilt | 1 | - | SGB 3 |
| **P[c]** | P-type | [c] | Basal | <a> | Twist | 1-3 | SGB 8 | - |
| **P[a]** | P-type | <a> | Prim Prism {1-100} | [c] | Tilt | 5-7 | | SGB 7,9 |
| **P[a]** | P-type | <1-100> | Sec Prism {1-210} | [c] | Tilt | 6-10 | | - |
| **P[a]** | P-type | <a> or <hki0> | Prism | <c+a> | Tilt | 29-37 | SGB 5,6,7 | SGB 1,11 |



**Table 2: Summary of the EBSD data statistics for EDML and NEEM samples. Columns show the SGB rotation axes, R, and rows the alignment of the SGB traces with respect to the basal plane shown in Figure 4. n is the number of SGBs, <c> is <0001>, <a> is <1-210>, <m> is <10-10>, * signifies dispersed rotation axes. The outermost 3 columns give simplified statistics of the same data set for comparison with X-ray Laue diffraction (Weikusat et al. 2011).**

| Rotation axis / SGB trace with basal plane | | <c> | <a> | <01-10> | <hki0> | <h0-il> | * | | <c> | in basal plane | * |
|---|---|---|---|---|---|---|---|---|---|---|---|
| **EDML** (total $n_{>0.5°}$=227) | | | | | | | | | | | |
| **N-type** | | 3 | 11 | 21 | 30 | 2 | 16 | | $3_{N[c]}$ | $62_{N[a]}$ | 18 |
| **P-type** | | 7 | $12_{P[a]prim}$ | $14_{P[a]sec}$ | 40 | 2 | 16 | | $7_{P[c]}$ | $66_{P[a]}$ | 18 |
| **other** | | 7 | 10 | 5 | 15 | 0 | 16 | | 7 | 30 | 16 |
| **NEEM** (total $n_{>0.5°}$=181) | | | | | | | | | | | |
| **N-type** | | 2 | 20 | 23 | 33 | 0 | 13 | | $2_{N[c]}$ | $76_{N[a]}$ | 13 |
| **P-type** | | 2 | $13_{P[a]prim}$ | $19_{P[a]sec}$ | 36 | 0 | 5 | | $2_{P[c]}$ | $68_{P[a]}$ | 5 |
| **other** | | 0 | 1 | 3 | 3 | 0 | 8 | | 0 | 7 | 8 |

**Figure Captions**

**Figure 1:** (**a**) Light microscopy (LM) image of etched surface of EDML ice (depth 694 m). Black arrow points to the top of the ice core. White arrow shows the trace of the c-axis. Large dark features are air bubbles in the ice. Features that are beneath the surface are out of focus and indistinct. GB = grain boundary, SGB = subgrain boundary. (**b**) Schematic diagram showing the orientation of sample sections in the ice core. Black arrows in (**a**) and (**b**) point towards the upper surface of the ice sheet. (**c**) Schematic diagram of the four types of SGB described in the text (after Weikusat et al. (2009).

**Figure 2:** Microstructures used for analysis of SGB types and slip systems: EDML sample (655.9 m depth). (**a**) LM image of map of sublimation-etched surface. Black arrow points to the top of the ice core. White arrow indicates the orientation of the c-axis. A P- and an N-type boundary are labelled. Red arrows show SGB in LM that are not observed in corresponding EBSD map. (**b**) EBSD map of the same area showing subsets of SGBs, labelled 1-12. (**c**) Pole figure of the central grain in the EBSD map showing c- and a-axes and schematic of the crystal orientation. (**d**) Rotation axes of neighbouring pixels across the SGBs 1-12.

**Figure 3:** Microstructure analysis from NEEM sample. (**a**) LM microstructure map of sublimation etched surface showing several GBs and SGBs. Vertical axis of the ice core is given by a dark grey arrow. White arrows indicate the orientation of the c-axes. A P- and an N-



type boundary are labelled. Red arrows show SGB in LM that are not observed in corresponding EBSD map. (**b**) Corresponding EBSD map showing SBG subsets labelled 1-12 and a schematic of the crystal orientation in the three grains shown. (**c**) Pole figure of the three grains shown in the EBSD map showing c- and a-axes with the colour coding as in (**b**). (**d**) Rotation axes of the SGBs 1-12.

5   **Figure 4:** Schematic diagram showing some SGB types (modified from Weikusat et al. 2010, 2011). Four end member types of boundary are shown. Rotation axes, R, are marked as circles with an X inside; parallel lines and hexagons represent the basal plane and are perpendicular to the c axis, shown by arrows or as a grey circle with a dot inside; dashed lines denote the SGB plane trace; in (**c**) the SGB plane is in the plane of the paper. (**a**) and (**b**) show SGBs with the boundary plane is perpendicular to the basal plane, that is N-type SGBs; in (**c**) and (**d**) the subgrain boundary planes are parallel to the basal plane, that is P-type SGBs. R is perpendicular to the c-axis in (**a**) and

10  (**d**) and parallel to the c-axis in (**b**) and (**c**). (**e**) shows a simplified inverse pole figure of the rotation axes parallel and perpendicular to the c-axis (dark grey circles). Possible slip systems for these SGB configurations are described in the text.

**Figure 5:** Schematic drawing of dissociation of screw dislocation b=<c+a> as one possibility to form a P[a] subgrain boundary. (**a**) <c+a> screw dislocation extending into an edge dislocation dipole (after Hondoh 2006). (**b**) Dissociation into b=[c] and <a> dislocations. (**c**) b=<a> dislocations glide away, b=[c] dislocations left behind to form P-type SGB.



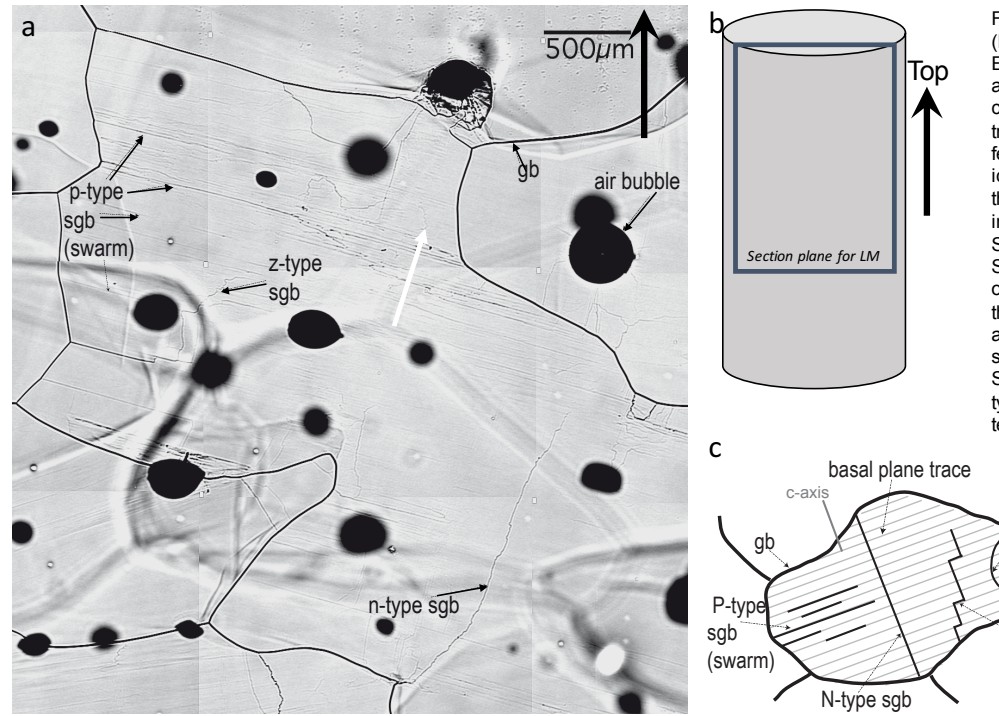

Figure 1: (**a**) Light microscopy (LM) image of etched surface of EDML ice (depth 694 m). Black arrow points to the top of the ice core. White arrow shows the trace of the c-axis. Large dark features are air bubbles in the ice. Features that are beneath the surface are out of focus and indistinct. GB = grain boundary, SGB = subgrain boundary. (**b**) Schematic diagram showing the orientation of sample sections in the ice core. Black arrows in (**a**) and (**b**) point towards the upper surface of the ice sheet. (**c**) Schematic diagram of the four types of SGB described in the text (after Weikusat et al. (2009).



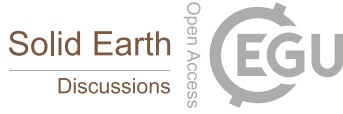

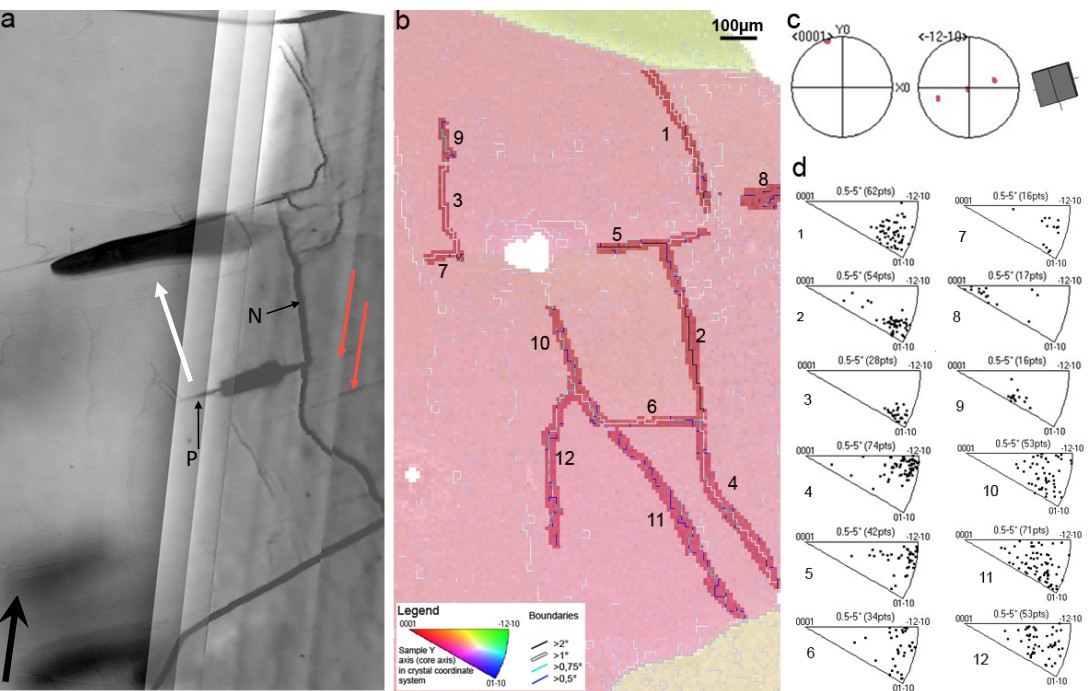

Figure 2: Microstructures used for analysis of SGB types and slip systems: EDML sample (655.9 m depth). (a) LM image of map of sublimation-etched surface. Black arrow points to the top of the ice core. White arrow indicates the orientation of the c-axis. A P- and an N-type boundary are labelled. Red arrows show SGB in LM that are not observed in corresponding EBSD map. (b) EBSD map of the same area showing subsets of SGBs, labelled 1-12. (c) Pole figure of the central grain in the EBSD map showing c- and a-axes and schematic of the crystal orientation. (d) Rotation axes of neighbouring pixels across the SGBs 1-12.



Figure 3: Microstructure analysis from NEEM sample. (a) LM microstructure map of sublimation etched surface showing several GBs and SGBs. Vertical axis of the ice core is given by a dark grey arrow. White arrows indicate the orientation of the c-axes. A P- and an N-type boundary are labelled. Red arrows show SGB in LM that are not observed in corresponding EBSD map. (b) Corresponding EBSD map showing SBG subsets labelled 1-12 and a schematic of the crystal orientation in the three grains shown. (c) Pole figure of the three grains shown in the EBSD map showing c- and a-axes with the colour coding as in (b). (d) Rotation axes of the SGBs 1-12.

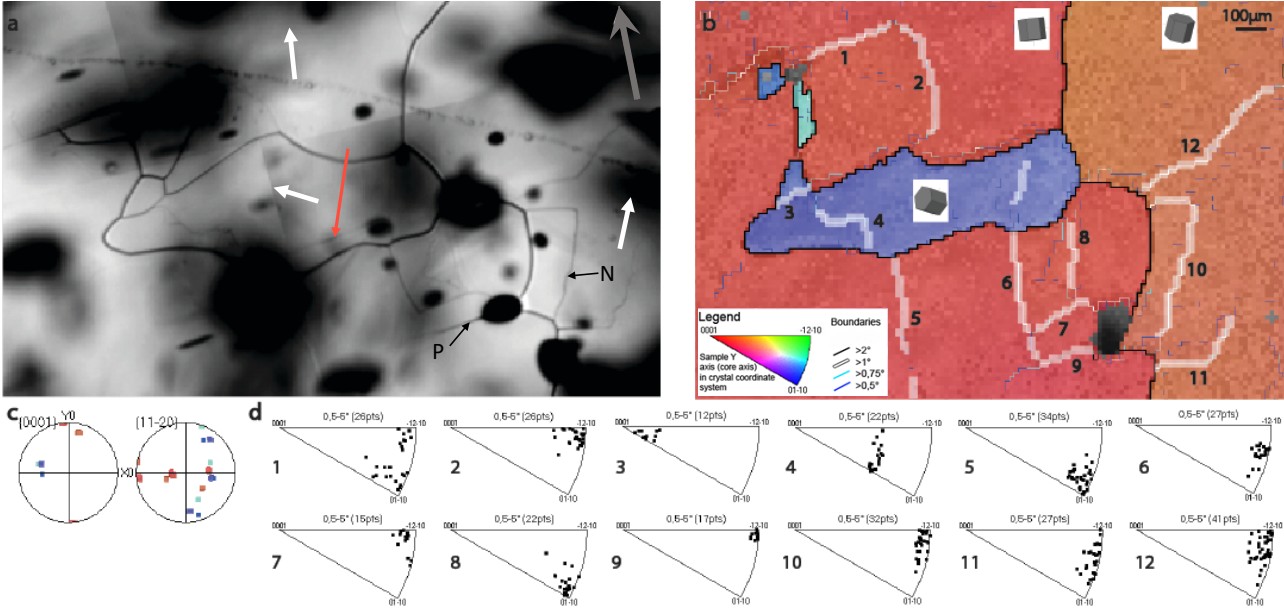



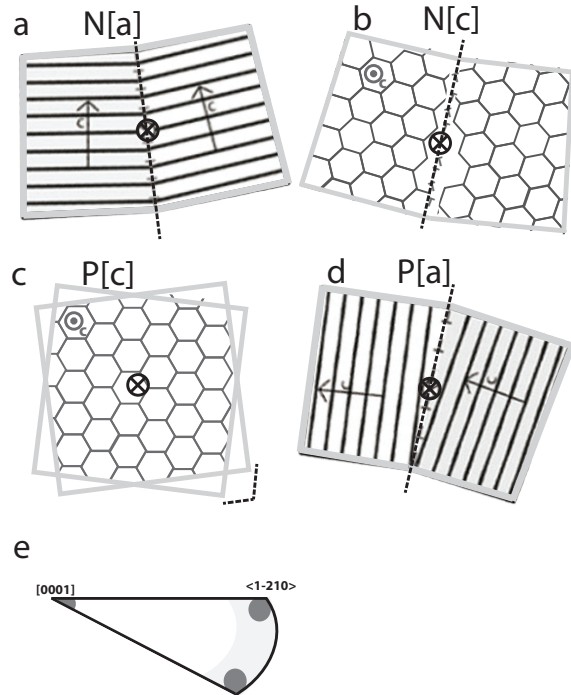

Figure 4: Schematic diagram showing some SGB types (modified from Weikusat et al. 2010, 2011). Four end member types of boundary are shown. Rotation axes, R, are marked as circles with an X inside; parallel lines and hexagons represent the basal plane and are perpendicular to the c axis, shown by arrows or as a grey circle with a dot inside; dashed lines denote the SGB plane trace; in (c) the SGB plane is in the plane of the paper. (a) and (b) show SGBs with the boundary plane is perpendicular to the basal plane, that is N-type SGBs; in (c) and (d) the subgrain boundary planes are parallel to the basal plane, that is P-type SGBs. R is perpendicular to the c-axis in (a) and (d) and parallel to the c-axis in (b) and (c). (e) shows a simplified inverse pole figure of the rotation axes parallel and perpendicular to the c-axis (dark grey circles). Possible slip systems for these SGB configurations are described in the text.





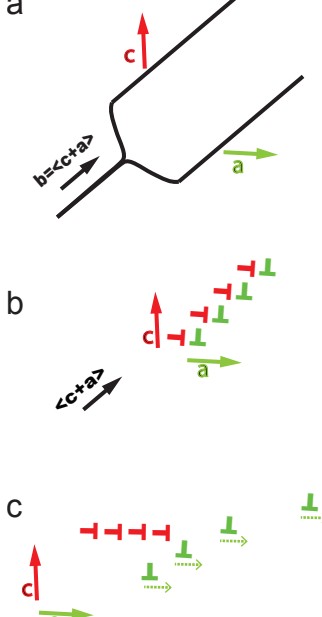

Figure 5: Schematic drawing of dissociation of screw dislocation b=<c+a> as one possibility to form a P[a] subgrain boundary. (a) <c+a> screw dislocation extending into an edge dislocation dipole (after Hondoh 2006). (b) Dissociation into b=[c] and <a> dislocations. (c) b=<a> dislocations glide away, b=[c] dislocations left behind to form P-type SGB.