# Peer review of "EBSD in Antarctic and Greenland Ice"

_Solid Earth, 2017_

## Short Comment (SC1) · 15 Mar 2017

First of all I would like to thank the author for their contribution to the important question concerning the type of dislocations involved during ice deformation. The question of non-basal dislocations is important to better constrain crystal plasticity law used in full field modelling using FFT (Llorens et al 2016, Suquet et al. 2011) or FE method (Richeton 2017). The evidence on non-basal dislocation has been recently highlighted by the previous work of the author using the similar tools (Weikusat et al. 2011) or by Piazolo et al, 2015 using Nye calculation through Weighted Burgers Vector.

My main concern is about the use of IPF (inverse pole figure) to determine the rotation axis trough the SGB (sub-grain boundary). It can lead sometimes to misinterpretation. For example concerning the N[a] SGB define as "N[a] describes a SGB trace normal to the basal plane (N-type) with rotation axis in the basal plane" (line 9, page 9). These SGB are interpret later as made of "Burgers vector b=<a> edge dislocations". But if the

information given by the IPF is that the rotation is lying in the basal plane, one cannot conclude directly whether the rotation axis is located within the SGB or perpendicular to the SGB (see figure), which will lead to different conclusions concerning the dislocations invokes. I would therefore suggest to the author to show the result in PF instead of IPF to avoid any ambiguity and misinterpretation.
* * *
[Figure]

[Figure]

Basal plane
• c axis
Boundary trace N
↻Possible rotation axis

**Fig. 1.**

---

## Referee Comment (RC1) · Anonymous Referee #1 · 18 Mar 2017

The manuscript "EBSD in Antarctic and Greenland Ice" submitted by Weikusat et al. for publication in SE represents a valuable contribution towards a better understanding of the flow behavior of polar ice sheets. The flow behavior of ice is an important topic in view of all glacial and sub-glacial Earth surface processes. The detailed and sophisticated light microscopic and EBSD data are well documented and the interpretations are comprehensive.

General comments are listed as follows:

1. The main observation of the study is a high amount of subgrain boundaries made up of dislocations representing non-basal glide systems. This observation is not entirely new but the study confirms their common occurrence in natural polar ice sheets.

Yet, the implication that this will "have a major impact on the discussion of strain-rate controlling processes" (abstract line 15, conclusions, page 15, lines 6-7) seems to be

overemphasized, as suggested by their discussion on page 14, lines 4-17: Even a minor activity of non-basal glide produces a high amount of relative immobile dislocations that are arranged into low angle grain boundaries. Thus, low angle grain boundaries made up of dislocations gliding on non-basal planes do not represent rate-controlling glide systems.

2. The glide system responsible for the main strain can most probably be best inferred from the CPO, yet the CPO is not documented in this study. It would be helpful to describe it, although it is not strongly developed, as stated on page 5, lines 22-24.

3. Generally, some more observations and discussions indicating that indeed deformation by dislocation glide is the main deformation mechanism of the studied ice samples would be helpful.

Specific comments are listed as follows:

- Page 1, line 15, rephrase, see general comment 1.

- Page 1 lines 16/17: "host grain alternative formation processes" please specify, which are these alternative formation processes (e.g., alternative to what. . ., what are the alternatives. . .).

- Page 2, line 12: "various deformation mechanisms": please specify the various deformation mechanisms

- Page 2, line 15, please specify: what is the main evidence for the interpretation that dislocation glide is the main deformation mechanism

- Page 3, line 22 (and throughout the text): please add: low misorientation "angles" < 5°. . .

- Page 3, line 25: please rephrase, e.g. . . . "the orientation/azimuth of" the c-axis of the ice is known. . ..

- Page 5, lines 22-25: Please add some information on the CPO, as this is important

to judge the importance of dislocation glide as main deformation mechanism

- Page 12, line 10 and following: please rearrange the listing of the seven types of subgrain boundaries recognized by Means and Ree, 1988. For example, give in the beginning of this discussion a short overview what are the main differences.

- Page 15, line 6, rephrase, see general comment 1.

- Page 5, line 11: strong CPO? Not in your sample?

References - Missing references in the list: Means and Ree, 1988; Mainprice et al., 1933

- Reference list: the year is at various positions in the listed references

- Please also cite the experimental work on the deformation behavior of ice by Piazolo et al., 2013; Cyprych et al., 2016

Figures

- Fig. 1: What would be the appearance of cleavage fractures in ice? Figure 1a) illustrates the "p-type sgb swarm". Given the low misorientation angle involved with these planar microstructures, could they represent cleavage fractures?

---

## Referee Comment (RC2) · Anonymous Referee #2 · 29 Mar 2017

This manuscript gives original results on the EBSD observations of sub- (or small angle) grain boundaries (SGB's) in deep ice cores recovered from Antarctica (EPICA-DML) and Greenland (NEEM). Introducing a new tool (EBSD) to determine small changes in crystallographic orientations within crystal grains, a number of different SGB's were successfully analyzed, resulting in a higher reliability of the statistics. The statistical distribution of different types of SGB's shown in Tables 1 and 2 is the main result of this study.

Table 2 clearly shows that both N(a) (i.e., tilt SGB's normal to the basal plane with rotation axes parallel to the basal plane) and P(a) (i.e., tilt SGB's parallel to the basal plane with rotation axes parallel to the basal plane) are predominant over other types, with almost equal probability for N(a) and P(a). The authors claim that this result strongly suggests both the basal and non-basal slip systems work in deformation of ice in deep ice sheets because the SGB's of N(a) and P(a) types are composed of the dislocations

with Burgers vector <a> and <c> (and/or <c+a>), respectively. Although the similar argument has been proposed in their previous papers, the present paper confirms the argument by highly reliable data obtained by the EBSD method. By this confirmation, the ice sheet flow modeling can be developed or improved on the basis of a reliable deformation mechanism of ice. Therefore, I recommend publication of this paper in Journal SE after some major and minor revisions by considering the following comments.

(1) Brush up the description in 'Abstract' by focusing on the main result. For example, there is a significant gap in author's argument in the description 'The finding that ........, is surprising. These findings are .... with respect sea-level evolution' (Line 13 to 16 on p.1), resulting in a confusion about the main topic of this paper.

(2) Line 12 on p.1: Is 'prism <a> or prism <c+a> slip' correct? Do you think no pyramidal slip is involved?

(3) The 'Introduction' seems to be too long with some redundant duplicates. Focus on introductory remarks (or reviews on closely related topics) required for understanding the arguments in the main text. For example, as far as I understand, the most important points that should be described in the introduction are (a) we have no direct evidence for the non-basal slips in glacier ice until now in spite of the shortage of independent slip systems to deform ice only by the basal slip system, (b) the evidence for non-basal slips can be obtained by finding SGB's which can be formed only when non-basal slips take place, and (c) a rapid SGB analysis becomes possible with the use of EBSD for the SGB analyses although number of data was not sufficient in previous studies because of the time-consuming method (x-ray Laue method). Other topics can be included in the introduction, but it should be concise as possible.

(4) Section 4.2 ; As the authors conclude the P(a) type SGB's are formed as a result of the non-basal slips in deformation of ice, the explanation for P(a) should be given more precisely. For example, the term 'non-basal edge dislocations' in line 22 includes

an ambiguity in 'non-basal' because it does mean either an edge dislocation lying on non-basal planes or an edge dislocation with Burgers vector lying on non-basal planes. Of course, the authors mean the latter, but it becomes clearer if it is written 'an array of edge dislocations with Burgers vector <c> or <c+a>'. Then, readers can easily follow the argument.

(5) What does 'controlled' in the title of Section 4.2 mean? Is any other word(s) appropriate here?

(6) The discussion given in the paragraph from line 28 on p.11 to line 6 on p.12 seems to fail to explain the dislocation processes involved in formation of P(a) type SGB's. Reconsider the description here. For example, 'limited strain can result in the formation of N-type tilt wall made up of immobile [c] dislocations' (Line 1 to 2 on p.12) is not understandable. More careful explanation is needed here without a gap in the argument. In addition, the closely related arguments are also given in Section 4.4. These should be combined.

(7) Seven types of SGB's in some rock-forming minerals are briefly described in the next paragraph and Sections 4.3 and 4.4, but with no explanations on a relation of different types of SGB's between ice and the rock-forming minerals. Is it possible to describe the definition of the seven types in rock-forming minerals with respect to the four types of ice given in this paper?

(8) What is the purpose of Section 4.4? It includes a formation mechanism of P(a) type SGB's which must be the main topic of this paper. As suggested above, this part should be combined with the closely related part in Section 4.2.

(9) Line 1 on p.14. 'X-ray tomography' should be 'x-ray topography'.

---

## Author Comment (AC1) · 5 Apr 2017

Dear Dr. Chauve, Thank you for your comment and figure. We agree with your concern, especially with respect to the N[a] type SGBs. Plotting the rotation axis in a pole figure, in specimen co-ordinates is a useful way to present the data, in addition to the inverse pole figure. We agree that N[a] type SGBs can be tilt or twist boundaries. Furthermore, N[a] twist boundaries will have a different dislocation structure compared to the N[a] tilt boundaries formed by an array of b=<a> dislocations. However, since we only have information on the SGB trace and lack information on the 3D orientation of the SGB plane, it is not possible in conventional EBSD on bulk samples to determine the relationship between the rotation axis and the SGB plane. In our interpretation we did assume that most N[a] boundaries are tilt walls, formed by basal slip, as expected from previous work (Hondoh 2000, 2010; Piazolo EA 2008). We will include your comment in the revised paper and refine the discussion to include the possibility of more complex

N[a] boundaries. The main message of our paper, that non-basal dislocations are more common than usually assumed from macroscopic behaviour, will remain the same.

Best regards Ilka Weikusat and co-authors